# Phylodynamics of SARS-CoV-2 Lineages B.1.1.7, B.1.1.529 and B.1.617.2 in Nigeria Suggests Divergent Evolutionary Trajectories

**DOI:** 10.3390/pathogens14111091

**Published:** 2025-10-26

**Authors:** Babatunde O. Motayo, Olukunle O. Oluwasemowo, Anyebe B. Onoja, Paul A. Akinduti, Adedayo O. Faneye

**Affiliations:** 1Department of Medical Microbiology, Federal Medical Centre, Abeokuta 110222, Nigeria; 2Biosciences and Biotechnology Division, Physical and Life Sciences Directorate, Lawrence Livermore National Laboratory, P.O Box 808, L-452, Livermore, CA 94550, USA; 3Department of Virology, College of Medicine, University of Ibadan, Ibadan 211101, Nigeria; 4Department of Medical Laboratory Science, Babcock University, Ilishan 121103, Nigeria

**Keywords:** virus evolution, SARS-CoV-2, VOCs, Nigeria

## Abstract

Background: The early months of the COVID-19 pandemic were characterized by high transmission rates and mortality, compounded by the emergence of multiple SARS-CoV-2 lineages, including Variants of Concern (VOCs). This study investigates the phylodynamic and spatio-temporal trends of VOCs during the peak of the pandemic in Nigeria. Methods: Whole-genome sequencing (WGS) data from three major VOCs circulating in Nigeria, B.1.1.7 (Alpha), B.1.617.2 (Delta), and B.1.1.529 (Omicron), were analyzed using tools such as Nextclade, R Studio v 4.2.3, and BEAST X v 10.5.0. The spatial distribution, evolutionary history, viral ancestral introductions, and geographic dispersal patterns were characterized. Results: Three major lineages following WHO nomenclature were identified: Alpha, Delta, and Omicron. The Delta variant exhibited the widest geographic spread, detected in 14 states, while the Alpha variant was the least distributed, identified in only eight states but present across most epidemiological weeks studied. Evolutionary rates varied slightly, with Alpha exhibiting the slowest rate (2.66 × 10^−4^ substitutions/site/year). Viral population analyses showed distinct patterns: Omicron sustained elevated population growth over time, while Delta declined after initial expansion. The earliest Times to Most Recent Common Ancestor (TMRCA) were consistent with the earliest outbreaks of SARS-CoV-2 globally. Geographic transmission analysis indicated a predominant coastal-to-inland spread for all variants, with Omicron showing the most diffuse dispersal, highlighting commercial routes as significant drivers of viral diffusion. Conclusion: The SARS-CoV-2 epidemic in Nigeria was characterized by multiple variant introductions and a dominant coastal-to-inland spread, emphasizing that despite lockdown measures, commercial trade routes played a critical role in viral dissemination. These findings provide insights into pandemic control strategies and future outbreak preparedness.

## 1. Introduction

The coronavirus disease 2019 (COVID-19), caused by SARS-CoV-2, was first reported in Wuhan, China, in December 2019 [1,2]. As of 27 August 2023, there were over 770 million confirmed cases and approximately 7.1 million deaths globally [3]. Africa has reported over 12 million infections and 250,000 deaths, with Nigeria recording over 265,000 cases [4]. Despite Nigeria’s large population, the country’s incidence rate (129 per 100,000 persons) remains relatively low compared to countries such as the United Kingdom and Indonesia, which reported significantly higher fatality rates [4].

Several hypotheses have been proposed to explain this disparity, including environmental influences, genetic factors, and possible pre-existing immunity due to prior exposure to related coronaviruses [5,6]. The role of molecular surveillance has been pivotal in tracking viral evolution, informing vaccine development, and monitoring emerging variants [7]. Since the publication of the first SARS-CoV-2 genome in January 2020 [8], over 3 million sequences have been shared globally, enabling comprehensive phylogenetic analyses [9,10].

In Nigeria, more than 55 SARS-CoV-2 lineages have been identified since the pandemic began [11]. While many are classified as Variants Under Observation (VUOs) or Variants of Interest (VOIs), several VOCs including Alpha, Delta, and Omicron “World health Organization WHO naming nomenclature; B.1.1.7, B.1.617.2, and B.1.1.529 “PANGOLIN nomenclature”. Collectively these have been associated with increased transmissibility and clinical severity [9,10]. This study aims to provide a detailed phylodynamic analysis of the major SARS-CoV-2 VOCs that circulated in Nigeria, focusing on their evolutionary rates, geographic spread, and transmission dynamics during the early waves of the pandemic.

## 2. Methods

### 2.1. Data Collection

Whole genome sequence data of SARS-CoV-2 variants from Nigeria were extracted from the global initiative for the sharing of Influenza virus database (GISAID) (https://gisaid.org/); assessed on 14 January 2023. Search queries included location, collection date, clade, and lineage. Variants B.1.1.7 (Alpha), B.1.525.2 (Eta), and B.1.1.529 (Omicron) were selected based on their relative abundance. Sample locations were exported in CSV format.

Samples were collected across various Nigerian states during the four COVID-19 pandemic waves. The Nigerian Centre for Disease Control (NCDC) coordinated the testing of clinical samples in designated laboratories in virtually all the states. Samples that tested positive to SARS-CoV-2 were sequenced at the African Centre for Excellence in Genomics of Infectious Diseases (ACEGID), Redeemer’s University, Ede, Nigeria; and the NCDC laboratory in Abuja, using Oxford Nanopore and Illumina MiSeq platforms, following established protocols [4,12]. The sequence data and associated metadata including the submitting laboratories are available in Appendix A.

### 2.2. Data Analysis

Lineage assignments were determined using Nexclade (https://clades.nextstrain.org/ accessed on 19 January 2023). Relative lineage distribution over time was analyzed in R Studio (R Core Team).

### 2.3. Phylogenetic Analysis

Genomic sequences of each VOC were aligned to the Wuhan-Hu-1/2019 (MN908947) reference genome using Nexclade [13]. Variant calling, phylogenetic placement, and clade assignments were performed automatically. Maximum likelihood phylogenetic trees were generated and visualized via the Nexclade web interface.

### 2.4. Phylogeography and Temporal Analysis

Evolutionary and temporal analyses were conducted using a Bayesian Markov Chain Monte Carlo (MCMC) approach in BEAST X v 10.5.0 [14,15]. Nucleotide alignments were generated from full high coverage sequences with MAFFT [16], with 206 Alpha, 89 Delta and 148 Omicron sequences. Temporal clock signal strength was evaluated using root-to-tip genetic distance with TempEst v1.5 [17]. Sequence details are available can be retrieved from GISAID with https://doi.org/10.55876/gis8.250911ba.

We partitioned the coding genes into first + second and third codon positions and applied a separate Hasegawa-Kishino-Yano substitution model with gamma-distributed (HKY + G) rate heterogeneity among sites to each partition. Two clock models were explored, including a strict and relaxed molecular clock, with four different tree priors, constant population size, exponential population size, Bayesian Skyride plot and Gausian Markov Random Field Skyride plot. We ran each model for an initial 30,000,000 states. Model comparison was carried out using Bayes factor test with marginal likelihood estimated and path sampling and stepping stone methods implemented in BEAST X v 10.5.0 [14,15]. The relaxed molecular clock model with a Gaussian Markov Random Field Skyride coalescent prior was eventually selected and applied, with 100 million MCMC states and a 10% burn-in with BEAGLE v 3.2 [18]. All effective sampling size ESS values were >200 to ensure sufficient sampling. Results were visualized using Tracer v1.8 (http://beast.community/tracer). Tree files were annotated in TreeAnnotator (http://beast.community/treeannotator) after a 10% burn-in, and visualized using ggtree [19]. Bayesian Skyline analysis was used to estimate viral population dynamics over time.

For phylogeographic analysis, geographic coordinates of sample locations were retrieved using google Earth (www.google.earth.com). A Bayesian stochastic search variable selection (BSSVS) model with Discrete traits were set in BEAST X v 10.5.0 [14,15] to infer geographic transmission routes at the state level. Continuous phylogeographic analysis was also conducted using a Cauchy relaxed random walk (RRW) model in BEAST X v 10.5.0 [14,15] using BEAGLE v 3.2 [18]. Annotated phylogeographic trees were generated using TreeAnnotator, after a 10% burn-in. The strength of migration routes of the various SARS-CoV-2 variants within their States of isolation were determined using Markov jumps and visualized as Chord plot in R-Studio (www.rproject.com). Continuous phylogeographic dispersal was visualized with SPREAD4 [20].

## 3. Results

### 3.1. Geographic Distribution of Variants

The geographic distribution of SARS-CoV-2 variants of concern (VOCs) in Nigeria is presented in Figure 1A. The Delta variant (B.1.617.2) exhibited the widest spread, detected in 14 states, while the Alpha variant (B.1.1.7) had the most limited spread, reported in 8 states.

### 3.2. Introduction and Spread over Time

Weekly variant introductions are shown in Figure 1B. The first three months of the pandemic were characterized by minimal variant introductions, with occasional detection of the Alpha variant. A sharp rise in Alpha variant introductions occurred between December 2020 and March 2021. The period from July to November 2021 experienced the highest frequency of multiple variant introductions. The Alpha variant displayed the highest duration of circulation during the study period, with three distinct epidemic waves. The highest multiple variant occurrence was observed in December 2021.

### 3.3. Phylogeny of Variants

Phylogenetic analysis of sequences from September 2020 to April 2022 revealed that the Omicron variant dominated, forming up to six sub-lineages (Figure 2). Delta variant sequences clustered into two sub-lineages (21J and 21I). Alpha variant sequences primarily aligned with their parental lineage. Additionally, a few sequences clustered with the Kappa (B.1.617.1) and Mu (B.1.621) variants.

### 3.4. Temporal Signal and Evolutionary Rates

Root-to-tip genetic distance analysis demonstrated the strongest temporal clock signal in the Delta variant (R^2^ = 0.05), followed by the Alpha (R^2^ = 0.31) and Omicron (R^2^ = 0.17) variants (Figure 3). Evolutionary rate estimates indicated that the Alpha variant had the slowest rate (2.66 × 10^−4^ substitutions/site/year), while the Delta variant evolved slightly faster (3.75 × 10^−4^ substitutions/site/year) (Table 1).

The most recent common ancestor (MRCA) for the Alpha variant was estimated to date back to August 2020, and the Delta variant’s MRCA emerged around May 2021 (Figure 4). These findings are consistent with the previously reported data, suggesting multiple introduction and circulation of major VOCs in different countries such as South Africa, England and USA [21,22,23].

### 3.5. Viral Population Dynamics

Bayesian Skyline analysis of effective viral population sizes over time showed that Alpha and Delta variants exhibited a steady population expansion. This is depicted by a gradual rise in Ne values over a period of a few weeks (Delta) to a few months (Alpha); as compared to over a year observed for Omicron (Figure 5). The Delta variant recorded a decline viral population shown by a decline in Ne towards the end of 2021. The Alpha and Omicron however both exhibited a steady Ne value towards the end of the study period.

### 3.6. Geographic Dispersal Patterns

The virus migration patterns of the 3 VOCs were analyzed using a discrete phylogeographic model. From the results of our analysis a circular migration plot was generated in form of a chord diagram, showing the inflow and outflow of each of the VOC between their States of detection (Figure 6). Our results show that the 3 VOCs showed similar geographic migration strengths, depicted by the width of their connecting chords between the inbound and outbound States.

Continuous phylogeographic analysis revealed consistent transmission trends for all three VOCs. Initial dispersal events originated from coastal southwestern Nigeria, predominantly Lagos, Ogun, Osun, and Ondo states. The Alpha variant exhibited localized spread within this region. In contrast, the Omicron variant displayed a more widespread dispersal, reaching northern states like Sokoto and Kano (Figure 7). The red dots on the maps represent the locations of the internal nodes of the phylogeographic tree, while the blue shaded regions represent 80% HPD uncertainty around the tree nodes in the maps.

Figure 6A–C show Circular migration flow charts, of SARS-CoV-2 variants Alpha, Delta, and Omicron, between States of identification in Nigeria. The connecting chords are based on Markov jumps from discrete MCC trees of the different SARS-CoV-2 variants. The outer ring shows outward migration, while the colors in the inner ring show inbound variant migration between the States.

## 4. Discussion

The COVID-19 pandemic had devastating global impacts, leading to overwhelming mortality rates, strained healthcare systems, and severe economic disruptions [24]. Nigeria was no exception, experiencing significant challenges despite implementing national and global control measures. Our study provides a comprehensive understanding of the genomic epidemiology of SARS-CoV-2 variants of concern (VOCs) in Nigeria. By analyzing whole genome sequences, we uncovered the patterns of viral introductions, geographic spread, evolutionary dynamics, and population expansion of key variants—Alpha (B.1.1.7), Eta (B.1.525.2), Delta (B.1.617.2), and Omicron (B.1.1.529). These findings are crucial for informing future public health strategies and pandemic preparedness.

### 4.1. Dominance and Geographic Spread of Delta and Omicron Variants

Our results highlight the Delta variant’s widespread geographic dispersal across 14 states, surpassing other VOCs in its reach. This expansive spread can be attributed to its enhanced transmissibility and immune evasion capabilities, as reported in previous studies [25,26]. The rapid expansion of Delta coincided with reduced public health measures, increased mobility, and vaccine delays in Nigeria. Similar patterns were observed globally, emphasizing the role of viral fitness and population susceptibility in shaping variant dominance.

Notably, the Omicron variant displayed remarkable adaptability, maintaining continuous circulation with multiple sub-lineages. This is consistent with global trends where Omicron’s higher immune escape potential led to widespread community transmission despite vaccination efforts [27,28]. In contrast, the Alpha variant was primarily localized to the southwestern states of Lagos, Ogun, Osun, and Ondo, suggesting a more restricted spread. The limited geographic footprint of Alpha, compared to Delta and Omicron, may indicate lower fitness or the effect of early containment measures.

### 4.2. Temporal Dynamics and Variant Introductions

The temporal analysis revealed a distinct timeline of variant introductions in Nigeria. The first detection of Alpha in October 2020 marked the initial wave of VOC emergence. Subsequent introductions of Delta and Omicron further amplified the viral burden. Our study suggests that these introductions were not isolated events but rather part of a continuous influx of new variants. Notably, the rapid rise in Delta introductions between July and November 2021 aligns with increased international travel and the relaxation of movement restrictions. Similar patterns have been observed in other countries with high human mobility [22,23].

Interestingly, our Bayesian evolutionary analysis suggests that the most recent common ancestor (MRCA) of the Alpha variant in Nigeria emerged around August 2020, preceding its first reported detection. This may indicate undetected community transmission before formal case identification, possibly due to limited testing capacity during the early pandemic months. While this finding should be interpreted cautiously due to potential biases in sampling size and sequencing coverage, it underscores the importance of robust genomic surveillance for early threat detection.

### 4.3. Evolutionary Rates and Population Expansion

The evolutionary rates of the VOCs provide valuable insights into viral adaptation. Alpha demonstrated the slowest evolutionary rate (2.66 × 10^−4^ subs/site/year), suggesting a relatively stable genome compared to Delta (3.75 × 10^−4^ subs/site/year). Omicron’s accelerated evolution, evident from its rapid accumulation of mutations, has been well-documented globally [27]. The continued dominance of Omicron sub-lineages highlights its remarkable adaptability, posing significant challenges to vaccination and treatment strategies.

Furthermore, the viral population expansion analysis revealed distinct patterns for each VOC. The Alpha and Omicron variants exhibited sustained population growth throughout the study period, indicating continuous transmission and adaptation. In contrast, the Delta variant experienced a rapid rise in viral population size around March 2021, peaking around June 2021 before slowly declining. This decline coincided with the rollout of vaccination campaigns and increased natural immunity, reflecting the effectiveness of population-level immunity in curbing Delta’s spread.

### 4.4. Phylogeography and Transmission Patterns

The phylogeographic analysis provided a clear narrative of the SARS-CoV-2 transmission dynamics in Nigeria. The viral migration rates were evaluated, the strength of viral migration between all States detected were similar as shown in the Markov jump chord plot (Figure 6). This observation shows a rapid dispersal of the variants between the States affected in Nigeria, regardless of their geolocation. All the VOCs showed a coastal-to-hinterland dispersal pattern, originating predominantly from Lagos and Ogun states, which serve as major commercial hubs. This northward and eastward spread along established trade routes is consistent with previous findings demonstrating the influence of human mobility on viral transmission [22,29]. The Delta variant’s extensive spread to northern states like Sokoto and Kano further underscores the role of economic and transportation networks in amplifying viral dissemination.

Despite travel restrictions, the continuous movement of people and goods likely facilitated the spread of VOCs. This reinforces the need for integrated molecular surveillance systems that can rapidly detect emerging variants and track their geographic spread. Implementing genomic sequencing capabilities at regional laboratories and establishing real-time data-sharing platforms will enhance Nigeria’s capacity to respond swiftly to future outbreaks.

### 4.5. Public Health Implications and Recommendations

The insights from this study underscore the critical role of genomic surveillance in pandemic management. Early detection of emerging variants, coupled with robust contact tracing and targeted interventions, is essential to contain outbreaks before they escalate. Additionally, our findings highlight the importance of adaptive public health policies that consider the dynamic nature of viral evolution.

To strengthen Nigeria’s pandemic preparedness, we recommend:Expansion of Genomic Surveillance: Establishing regional sequencing hubs and training local scientists will enhance the country’s capacity to monitor viral evolution and detect emerging variants in real time.Enhanced Data Integration: Developing a national genomic data-sharing platform will facilitate rapid dissemination of insights to inform public health decision-making.Targeted Public Health Measures: Building capacity for mobile testing units, quarantine stations, and digital contact tracing in high-traffic regions can mitigate the spread of fast-moving variants during future epidemics.Vaccination and Booster Campaigns: Prioritizing booster doses for vulnerable populations and maintaining vaccine coverage will reduce the risk of severe disease from emerging variants.International Collaboration: Continued collaboration with global genomic surveillance networks like GISAID will provide valuable insights into regional and global viral evolution.

## 5. Conclusions

This study provides a comprehensive genomic and phylogeographic analysis of SARS-CoV-2 VOCs in Nigeria, highlighting the complex interplay of viral evolution and public health responses. Our findings underscore the importance of sustained genomic surveillance, adaptive public health measures, and international cooperation to mitigate the impact of future pandemics. By applying these lessons, Nigeria and other nations can strengthen their preparedness for emerging infectious diseases and safeguard public health in the years to come.

## Figures and Tables

**Figure 1 pathogens-14-01091-f001:**
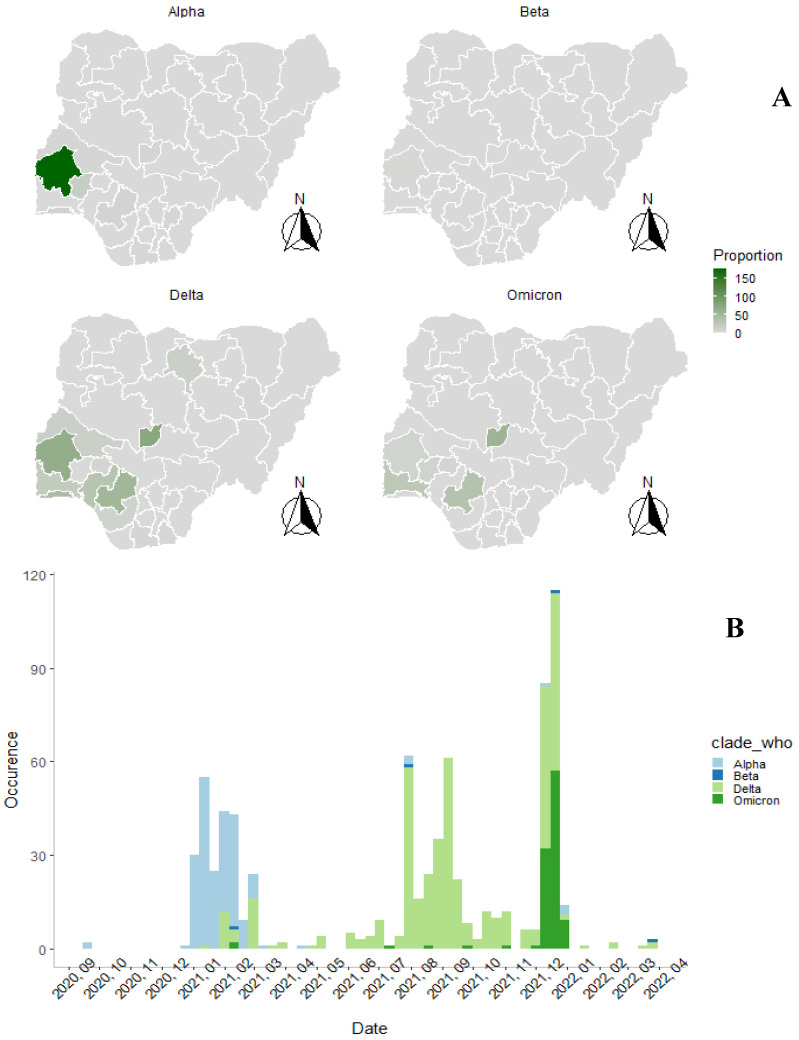
(**A**) Geographic distribution of the total number of sequences originating from patients in the various States of Nigeria. (**B**) Monthly distribution of SARS-CoV-2 Variant across Nigeria between August 2020 and April 2022, Legend indicates color coding for variants according to WHO guiding nomenclature.

**Figure 2 pathogens-14-01091-f002:**
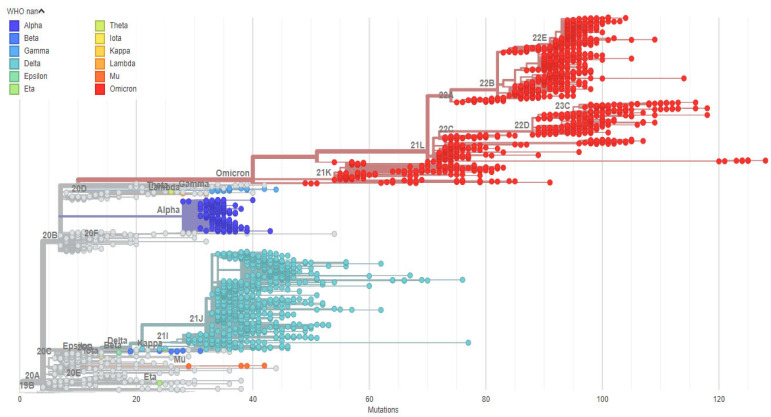
Phylogeny of SARS-CoV-2 Variants isolated in Nigeria 2020 to 2022. Tree was constructed using maximum likelihood GTR + I model and visualized using Nextrain. Legend shows the various clades identified using WHO-guided naming nomenclature.

**Figure 3 pathogens-14-01091-f003:**
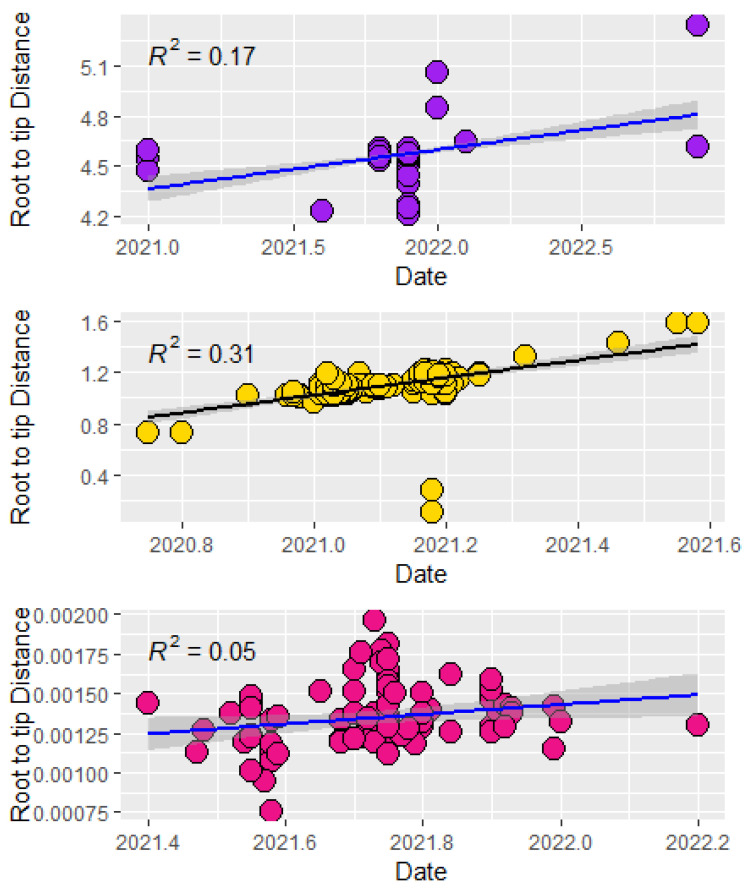
Root to Tip distance of nucleotide sequences of major circulating variants in Nigeria 2020 to 2022. Topmost: Omicron; middle: Alpha Variant; bottom: Delta Variant.

**Figure 4 pathogens-14-01091-f004:**
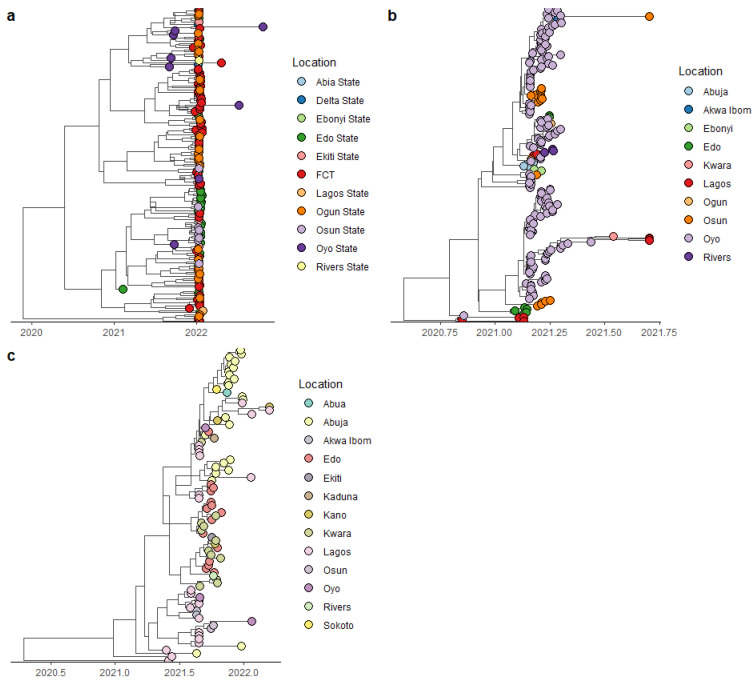
Time-scaled MCC trees of Nigerian SARS-CoV-2: (**a**) Omicron variant, (**b**) Alpha variant, and (**c**) Delta variant.

**Figure 5 pathogens-14-01091-f005:**
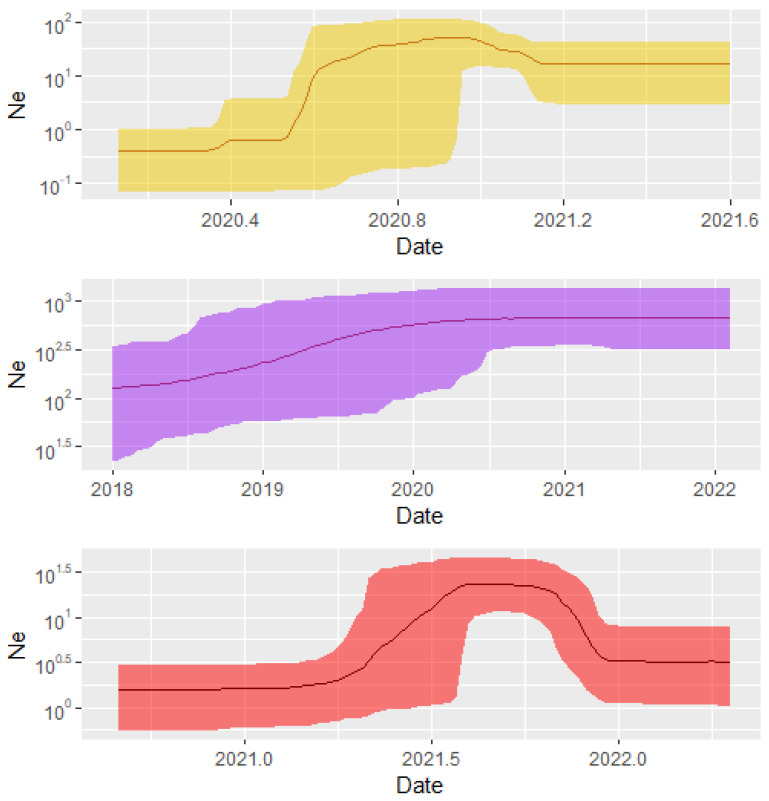
Bayesian Skygrid plot showing the viral population expansion through time of the main SARS-CoV-2 variants that circulated in Nigeria from 2020 to 2022. Topmost: Alpha Variant; middle: Omicron; bottom: Delta Variant.

**Figure 6 pathogens-14-01091-f006:**
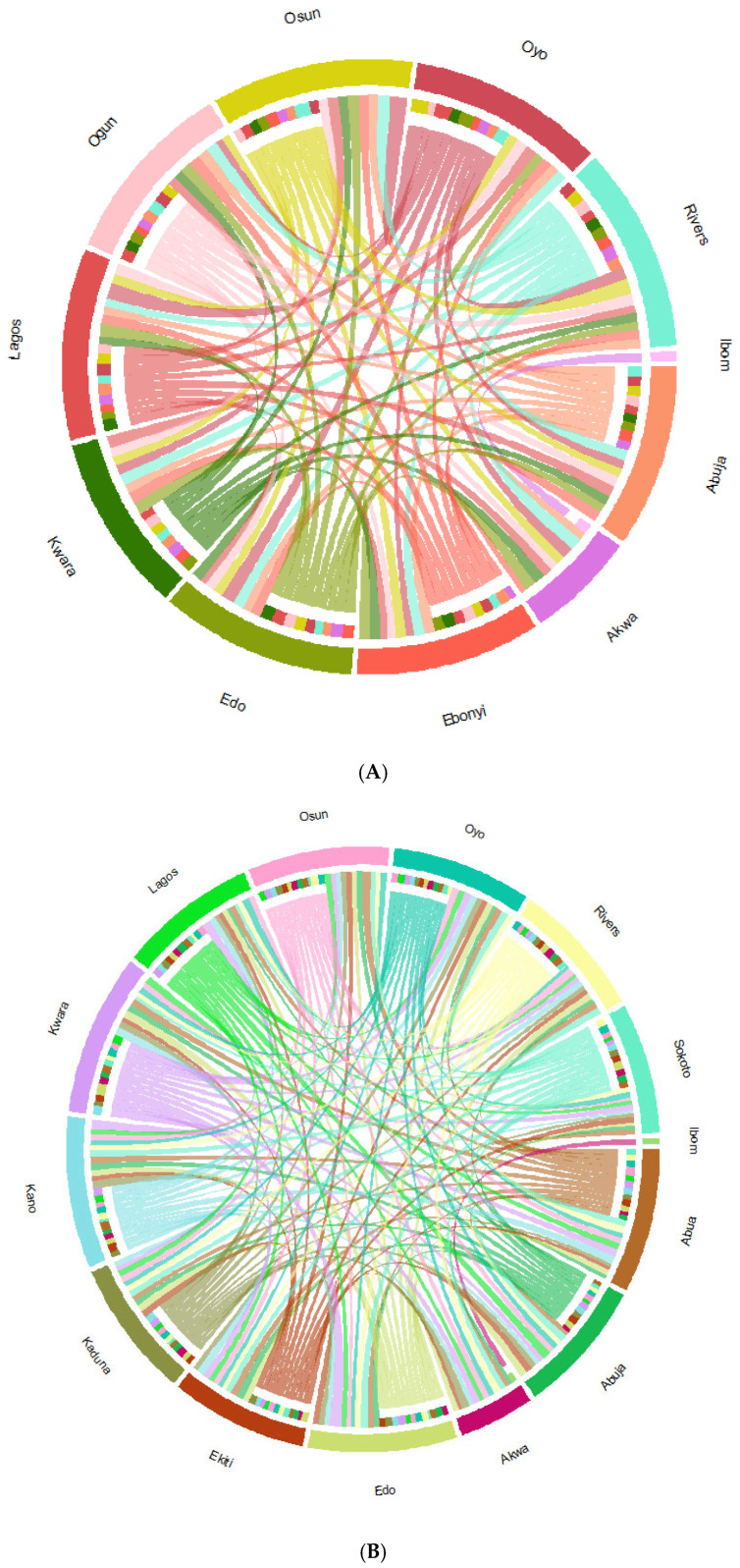
(**A**) Alpha; (**B**) Delta; (**C**) Omicron.

**Figure 7 pathogens-14-01091-f007:**
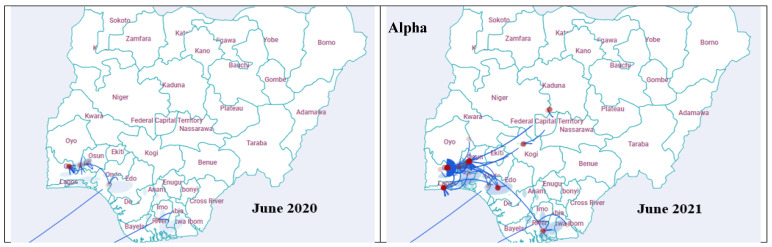
Continuous phylogeographic diffusion of SARCoV-2 lineages B.1.1.7 (Alpha variant), B.1.617.2 (Delta variant), B.1.1.529 (Omicron) isolated from different states in Nigeria during the study period. The red dots represent internal nodes of the phylogeographic tree, the blue connecting lines represent the VOC dispersal paths. The blue shaded regions represent 80%HPD around the tree nodes in the maps.

**Table 1 pathogens-14-01091-t001:** Evolutionary rate and time to most recent common ancestor (TMRCA) dates of Nigerian SARS-CoV-2 variants circulating in Nigeria.

Variant	Evol Rate	95% HPD Evol Rate	TMRCA	95% HPD TMRCA
Alpha	2.66 × 10^−4^	1.67 × 10^−4^ to 3.47 × 10^−4^	July 2020	Nov. 2018–Sept. 2020
Delta	3.75 × 10^−4^	4.17 × 10^−4^ to 7.71 × 10^−4^	May 2020	May 2019–Feb. 2020
Omicron	4.17 × 10^−4^	2.72 × 10^−4^ to 7.96 × 10^−4^	Dec. 2019	Sept. 2018–Oct. 2020

## Data Availability

The original data presented in the study are openly available in [GISAID] at Appendix A.

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
