# Peer review of "Phylodynamics of SARS-CoV-2 Lineages B.1.1.7, B.1.1.529 and B.1.617.2 in Nigeria Suggests Divergent Evolutionary Trajectories"

_pathogens, 2025, doi:10.3390/pathogens14111091_

Round 1

Reviewer 1 Report

Comments and Suggestions for Authors

All the comments to the manuscript are inserted in the attached document

Comments on the Quality of English Language

Quality of English Language needs to be improve

Author Response

Comment 1; The authors should check this information since in other part of the text mention another lineage

Response: We have checked and corrected the error in the lineage nomenclature.

Comment 2; Text needs to tbe improve: missing periods, incorrect prepositions, awkward constructions ("Sequence details are available can be retrieved")

Response: We have improved on the text and language quality.

Comment 3; The authors need to follow the standardized nomenclature (e.g., WHO, Pangolin, Nextstrain), and define terms early.

Abstract

Response: We appreciate the reviewer for the comment, we have revised the manuscript with the lineages following the WHO and Pangolin standardized naming nomenclature. We have also defined terms as requested.

Comment 4.; Please include period

Response: Done

Comment 5; Improve redaction

Response: Done

Comment 6: How many lineages were found?

Response: We have included the number of lineages found thank you.

Introduction

Comment 7; Why authors decide to use this date and no other more recent? Abstract give not context of this

Response: We have reviewed the date and reference thank you thank you.

Methods

Comment 8; The methods section need more transparency in order to make possible the reproducibility.

Response: We appreciate the reviewer for the comment, we have revised the methods section in line with the reviewer’s comment and have improved on the transparency thank you.

Comment 9; It is important to mention the date on which the authors acceced to the database

Response: Done

Comment 10; Different format

Response: Corrected

Comment 11; There is no mention over the number of sequences used in BEAST

Response: We have included the number of sequences from each Variant used in BEAST analysis.

Comment 12; How was the prios selection for the analysis?

Response: We thank the reviewer for this comment. Firstly, we tested each of the tree priors under both strict and relaxed molecular clocks, using the Bayes Factor method. Guided by relevant literature we chose the most optimum prior (GMRF Bayesian Skyride) for the final BEAST analysis. We have also included details of the selection strategy in the revised manuscript.

Comment 13; The author used any substitution model? ther is no mention on ESS values. Also, the author used any filter of quality thresholds?

Response: We thank the reviewer for the comments. We have included details of the substitution model used, as well as the cutoff ESS value of  >200.

Comment 14; Author should spicify this information

Response: Done

Results

Comment 15; Image needs to be improve.

Response: Image quality has been improved.

Comment 16; The authors mention as a strength of the study the fact that it has an impact on public policy. However, this study evaluates the dynamics of variants that are no longer circulating but have been replaced by others with different behavior, which renders decision-making with variants other than the current ones ineffective. The authors should included current lineages.

Response; We appreciate the comment and observation of the reviewer, however the scope and objective of the study is to derive insight from the most abundantly circulating variants of concern during the peak period of the COVID-19 pandemic Nigeria. Secondly to conduct proper evolutionary analysis at a national level, high number of these new variants are required. Most of which are too few in Nigeria to generate any useful information regarding their phylodynamic uniqueness or evolutionary pattern. Thirdly judging from the duration of circulation of some the VOC’s analyzed in our study ( over one year), insights derivable from such long-term circulation of a highly and rapidly diversifying pathogen such as SASCoV-2 VOC’s adds significant information which may guide deployment of epidemic intelligence tools such as was described in this study particularly in a resource poor environment like Nigeria.

Comment 17; The authors do not evaluate this variable

Response: We agree with reviewer and have deleted the phrase stating the analysis of the variable, thank you.

Comment 18; Reference need to be check it

Response: Done

Reviewer 2 Report

Comments and Suggestions for Authors

The authors investigated the evolution and migration of SARS-CoV-2 in Nigeria by computational analysis. The results show that the three major variants of concern (VOCs) migrated differently and dominated in various regions, which might correlate with their distinct phylogenic characters. Overall, this work provides some valuable information, but the quality of the data presentation is low, hampering the conclusions. Thus, I believe the manuscript requires a major revision and my specific suggestions/questions are followed:

Figure 1B is not clearly annotated. Which ones are Alpha, Delta and Omicron, respectively?

Why do the authors state there are four sub-lineages of Omicron? Figure 2 seems to show much more than four.

Some of the R2 values reported in the text are different from the numbers shown in Figure 3.

Figure 4C is confusing. What are the values (0, 25, 5, 75) following the years? Are they quarters of a year? If so, the MRCA of Alpha should not be July 2020. Also, this figure shows the MRCA of Delta appeared in May 2021 not May 2020, in contrast to the authors’ statement on Delta’s ‘rapid global spread…in the early months of the outbreak’.

How were ‘steady’ and ‘sharp’ defined in Figure 5? The plot of Alpha also shows a peak, but why is it interpreted as a ‘steady growth’?

Figure 6 is highly blurred, making the annotations unidentifiable.

Figure7 is not properly annotated. What do the red dots represent? What does the gradient blue indicate?

Author Response

Comm 1. Figure 1B is not clearly annotated. Which ones are Alpha, Delta and Omicron, respectively?

Response: We have edited Figure 1B and improved the annotated the bar graph to represent the different VOC’s as suggested by the reviewer

Comm 2. Why do the authors state there are four sub-lineages of Omicron? Figure 2 seems to show much more than four.

Response: We thank the reviewer for this observation, we have made the correction by stating the correct number of Omicron sub-lineages.

Comm 3. Some of the R2 values reported in the text are different from the numbers shown in Figure 3.

Response: We have made the correction on the R2 values in the main text.

Comm 4. Figure 4C is confusing. What are the values (0, 25, 5, 75) following the years? Are they quarters of a year? If so, the MRCA of Alpha should not be July 2020. Also, this figure shows the MRCA of Delta appeared in May 2021 not May 2020, in contrast to the authors’ statement on Delta’s ‘rapid global spread…in the early months of the outbreak’.

Response: We appreciate and agree with the reviewers observation and have re-casted the description of the MRCA particularly that of Alpha from July to August and Delta from May 2020 to May 2021. We have also re-casted the conflicting statement thus “These findings are consistent with the previously reported data, suggesting multiple introduction and circulation of major VOC,s in different countries such as South Africa, England and USA(Tegally et al, 2021; du Plessis et al, 2021; Zeller et al, 2021)”

Comm 5. How were ‘steady’ and ‘sharp’ defined in Figure 5? The plot of Alpha also shows a peak, but why is it interpreted as a ‘steady growth’?

Response: We thank the reviewer for the comment, we have since corrected the sharp rise and rise in population growth was defined as increase in Ne values against time, while steady was defined as stable Ne value over time along the course of the study duration. We have also revised the interpretation of the peak seen in Alpha as suggested by the reviewer.

Comm 6. Figure 6 is highly blurred, making the annotations unidentifiable.

Response: We have improved on the quality and format of the Figure.

Comm 7. Figure 7 is not properly annotated. What do the red dots represent? What does the gradient blue indicate?

Response: We have properly annotated the Figure describing what each feature represents both in the result text and the Figure legend.

Round 2

Reviewer 2 Report

Comments and Suggestions for Authors

I appreciate the authors for their efforts in improving this work. All my questions have been addressed in the revised manuscript, which is publish-ready.